# Online Reputation for Food Sector Exporters in the B2B Context: The Importance of Sustainability

**DOI:** 10.3390/foods12203862

**Published:** 2023-10-21

**Authors:** Manuel Jesús Puma Flores, Isabel María Rosa-Díaz

**Affiliations:** Department of Business Administration and Marketing, Faculty of Economics and Business Sciences, University of Seville, 41018 Sevilla, Spain; manpumflo@alum.us.es

**Keywords:** agri-food exports, online reputation, business to business, sentiment analysis, sustainability, certifications

## Abstract

Food exporting companies play a fundamental role in the development of international trade. However, the conceptualization and measurement of their online corporate reputation has not been extensively studied in Business to Business markets, unlike in Business to Consumer contexts. The aim of this research is to identify the variables that determine the online reputation of food companies operating in B2B markets, and to analyze their relationship with the volume of food exports. For this purpose, a three-stage quantitative and qualitative study has been developed, based on in-depth interviews with experts from export organizations and managers of Peruvian food exporting companies, the estimation of an Advanced Sentiment Analysis, the construction of a Total Online Reputation Index (TOR), and the development of regression analysis. The study has identified 13 variables that affect the online reputation of food exporting companies and indicates that the presence of sustainability content on their website and the number of pages visited positively affect the volume of food exports. Moreover, the TOR could have a significant explanatory capacity with respect to company exports. These results constitute a reference guide for both companies in the sector and official export agencies, highlighting the critical aspects to promote their exports.

## 1. Introduction

The current transition towards the digitalization of the commercialization of products and services has brought with it a series of methodologies and studies for measuring digital reputation and its relationship with the trust it generates in customers [1,2]. These investigations and methodologies have focused primarily on the Business to Consumer (B2C) context, with little development in the Business to Business (B2B) area, which has particular and differential characteristics and circumstances. It is precisely this knowledge gap that guides this study.

Among the different lines of research related to the digital and interactive marketing, the growing importance of online reputation in academia stands out, as a result of the priority given to it by companies [3]. Despite this, there is little consensus on the definition of online reputation and on its measurement [4]. Specifically, the conceptualization encompasses varied perspectives, which consider online reputation as a capability of the firm in the online context [5], the balance between the positive and negative impact of the firm’s digital commerce activities [6], an indicator of the quality of online marketers and information creators [7], the extent to which users can identify the position of others, including themselves, in a social networking environment [8], or a set of opinions of customers shared on websites about a product, service, or brand [9], with the ability to reduce information asymmetry [10].

Regarding the measures used with respect to online reputation, two main types of proposals can be found:The general assessments, such as the total number of downloads of a product in the online context [11], and the online consumer review ratings [12,13].The composite indices, which propose different online reputation constructs and components: online visibility, valence, volume and consistency of online signals, and perceived trustworthiness [11]; volume or the number counts of reputational ratings, valence, or the average of these ratings and seller profile ratings at the moment of the offer [10]; product information, conveyance, website content and ranking, offers and promotions, advocacy, status delivery, and privacy and security [14]; service quality, perceived value, and recommendation rate [9].

The digitalization of companies at the commercial level is of great importance for their internationalization, for their digital innovation, and for the analysis and execution of marketing strategies [15,16]. As a result, online reputation takes significant importance in the purchase and sale of products, especially in areas such as the food sector [17,18], with particular emphasis on security aspects [19]. This has led to the considerable development of online reputation research in B2C and Customer to Customer (C2C) platforms. However, online reputation has not been studied in depth and adapted to the B2B markets, particularly in the field of food exports [20,21]. Because of the different nature of B2C and B2B markets, there is a need to provide online reputation management tools adapted to each context [22,23]. This gap highlights the request and opportunity to develop an online reputation measurement system specific to the B2B food export domain. This issue represents the central aim of this research.

With the development of information and communication technologies (ICT), the online reputation of companies has become especially relevant in sectors as diverse as the financial markets [24], the global software industry [25], chemicals and materials, healthcare and diagnostics, automotive, and insurance [26]. This circumstance is not alien to the food sector [27], due to its extremely sensitive nature and impact on the international economic context [18,19]. Similarly, digitalization has eliminated barriers to trade, also allowing small and medium-sized enterprises (SMEs) to become more involved in international trade [28], thus increasing their food exports and, in turn, their need to clearly understand the factors that condition their online reputation. This is precisely the main utility of this study.

In addition to the above, the situation derived from COVID-19 has brought an opportunity and a need for online trade in the food field [29]. For the specific case of the promotion of food exports, during the COVID-19 crisis, such a relevant medium as B2B international food trade fairs went from a face-to-face format to a virtual or hybrid format. The most emblematic and successful fairs of the sector in Europe made this change to face this uncertain situation, evidencing the weakness of many companies to position themselves in the digital field efficiently [22,30]. This turns out to be crucial, since digital media presence, and in particular virtual exhibitions, constitute a fundamental platform for the commercial and marketing activities of companies [30], especially in this type of adverse conditions. This has made it crucial for companies to implement practices that transmit security and trust, both to client companies and to final consumers, in the digital food commerce environment [31,32].

In short, the transition towards an increasingly digital and sustainable food sector has led to the requirement for specific indicators linked to online reputation (digital reputation or e-reputation). In addition, detailed information is needed to properly manage the aspects that make up the online reputation of companies. This study aims to reduce this information gap in the B2B context of food exports with the following contributions:

First, from the business point of view, the development of marketing strategies for positioning and improving the image and reputation of food companies, especially in the case of new SMEs, leads to the need for an effective digital transition, with these digitalization processes being especially relevant for food exports [29]. The study conducted in this research presents the results of interviews with Peruvian food export companies involved in B2B transactions in Europe, identifying variables that influence online reputations in this context. This was also based on a previous literature review of relevant studies on B2B digital reputation. These results provide a specific practical reference for companies in the food sector, as well as a stimulus for the development of more in-depth studies in this area, which provide tools adapted to their particularities.

Second, marketing is evolving into a digital future, with the field of marketing analytics increasingly challenged to make sense of more data from diverse sources [27,33]. These data in turn represent an opportunity to provide a wider range of indicators to measure online reputation [34]. In this regard, this study diversifies sources of information, insofar as it uses quantitative and qualitative variables to measure online positioning, as will be described in the Materials and Methods section.

Third, there is a lack of research into specific aspects of online reputation in the B2B context, particularly in the food sector. Among them, this study has focused on company websites, social media, and e-marketplaces in B2B markets, which need to be managed differently than in B2C contexts, or in areas of blending between B2B and B2C strategies [17,23,35].

Finally, the COVID-19 crisis reinforced the importance of the digital reputation of companies (given the restrictions on face-to-face business transactions) and highlighted the need to strengthen the study on the digital transition in the context of B2B trade [29,30]. The scenario selected for this research has been international food trade shows, which are important vehicles for export promotion.

The paper is structured as follows. The next section develops the theoretical framework of the research leading to the formulation of the hypotheses of the study. This is followed by an explanation of the methods used and the results achieved, as well as a discussion of them. Finally, conclusions and implications for management are drawn, and future research is suggested.

## 2. Theoretical Background

Online reputation (e-reputation or digital reputation) can be defined as the perception built on a company, brand, product or service, based on the set of elements and information that circulate in online or digital channels, and which are linked to the mission and vision of the companies [36], their corporate social responsibility (CSR) actions [37], as well as the credibility of the information they provide or that is disseminated about them in digital contexts [2]. Therefore, the online reputation of a company represents an open process linked to its image and identity [38], in which customers share their evaluations and opinions about firms through the social media available on the Internet [34], forming an image of the company, product or service, and influencing customers’ shopping experience and purchasing decisions [39], as well as their loyalty [40,41].

This approach is based on the tenets of the theory of consumer behavior, which states that consumers make purchasing decisions after searching for information and comparing alternatives [42]. The main reason for this is that, as the theory of cognitive dissonance suggests, consumers try to minimize the cognitive dissonance (postpurchase psychological discomfort) that results from erroneous or inappropriate decisions related to the purchased product and to the emotional issues [43].

For the specific case addressed in this study, namely food exports in B2B online markets, reputation can become a key element for the purchasing decisions of client companies [18]. This approach is in line with the premise of the resource-based theory, which highlights the combination of resources and capabilities as antecedents of achieving competitive advantages [44]. Indeed, reputation affects confidence in the fulfilment of trade agreements [29], including aspects such as the quality and safety of food products [19] and the efficiency of the distribution systems [45]. Moreover, the purchasing decisions of food customers are not only based on performance and price criteria, but also on ethical and environmental issues, which are key to the reputation of companies [46].

Based on the above arguments, we put forward the following hypothesis:

**Hypothesis 1:** 
*Online reputation has a significant relationship with the volume of food exports of companies in the B2B context.*


In this context, taking as a reference the stakeholder theory and the central role of trust and credibility in the relations between companies and their customers [47], it is essential for food export companies to consider the variables that most decisively affect their online reputation. In the B2B online markets, customers are companies that evaluate the online information available. Therefore, the level of internet presence that companies have [39], as well as their positioning in internet search engines [48,49], transmit signals of notoriety, solidity, and even credibility and trust [2], which become decisive for their business and results.

The question that arises at this point is whether the phenomenon described occurs in a relevant way in B2B exports of food products. Few studies have been carried out in this respect [23,25], although their results provide a basis for considering that the internet presence of companies acting in B2B markets has a positive influence on their business performance, as is observed in B2C markets. It is therefore of interest to incorporate the question into this research, leading to the formulation of the next hypothesis:

**Hypothesis 2:** 
*The volume of publications related to the exporting company indexed on the internet has a significant relationship with the volume of exports.*


Another relevant issue is that consumers use information that they find understandable, relevant, and credible to them in their purchasing decisions [50]. Thus, the information they access is incorporated into their decisions, generating affective states that in turn lead to attitudes and behaviors. Moreover, the intensity with which these processes develop depends on the perceived informational value of the inputs [51], and especially on the sentiments conveyed by the published information [52,53]. Thus, publications in online environments that have positive connotations and are credible tend to produce favorable customer attitudes and behaviors, including purchase decisions and loyalty [40].

The relationship between the positive or negative news regarding the company’s food products (sentiment) and the export performance of food companies need to be studied in greater depth in this sector [54]. Based on the reasoning presented so far, the following hypotheses has been formulated:

**Hypothesis 3:** 
*The sentiment of food exporting companies’ publications indexed on the internet has a significant relationship with their export volume.*


Another key element in shaping a company’s online reputation involves the quality of its website, which refers to its accessibility, attractiveness, informative and visual quality, and the trust it generates [55]. Specifically, the experiences in the use of the selling company’s website leads users to build a perception and assessment, which ultimately contribute to the online reputation of that company [56], and consequently to its results in terms of food exports. Variables, such as the frequency of content update, the number of pages visited, and the quality of storytelling within the website, as well as translation into other languages, are related to the quality of a well-developed website [52,57].

In addition, in the specific case of food exporting companies in B2B markets, it is essential to add those elements of the website that transmit reliability, safety, and commitment [33], given the perishable nature of many food products, as well as their direct association with health and environmental protection [58]. As a consequence, public information on the social and environmental engagement of food exporting companies, as well as on the certifications obtained, potentially plays a strategic role in their exports.

The arguments put forward provide the basis for the following hypothesis:

**Hypothesis 4:** 
*The quality of the website, including sustainability information and certifications, has a positive relationship with the export volume of exporting companies in the B2B context.*


A further element of major relevance for the configuration of a company’s online reputation is its presence on social networks. In earlier times, purchasing decisions were based on consumers’ own knowledge and experiences, the comments of people close to them, and the promotional activities of companies. Nowadays, social networks exert a decisive influence on opinions, attitudes, and purchasing behaviors, creating a large number of connections and broad information bases that are shared by all participants [59]. Companies cannot remain oblivious to this reality and should consider that the attitudes they inspire in online communities [42,60], the engagement strategies they develop [61], and the people and social influencers with whom they collaborate [62,63], may exert a decisive influence on their business performance through their online reputation [51]. 

Evidence from previous research confirms the strong impact of social media on online reputation and consumer purchasing behavior in sectors such as textiles [50], tourism [41,64], and food and beverages [34], in B2C markets. The question that arises here is to what extent the presence of food exporting companies on social media can promote their export volume in B2B markets [35]. This leads to the following hypothesis:

**Hypothesis 5:** 
*Presence in social networks has a positive and significant relationship with food exports of companies in the B2B context.*


A final question addressed in this study is whether the participation in B2B marketplaces (e-commerce), that is, through Internet-based electronic marketplaces that enable and enhance online B2B relationships, can foster exports of food products.

E-commerce is proving to be a very favorable medium in the marketing of products in B2C markets, as it considerably increases the flow of information, as well as the possibility to compare product qualities and prices [41,64]. In the specific case of B2B contexts, marketplaces offer potentially advantageous scenarios for companies operating in them, since the volume of trade is considerably higher than in B2C markets, allowing cost savings and economies of scale [17]. 

At the same time, the particularity arises that substantial economic transactions may require more tangible contacts between the companies involved, in order to generate greater trust [44,65]. This is compounded by the specificities of food products, especially perishable ones such as agricultural products, which require more direct monitoring to ensure both their quality and their proper preservation [66]. 

A number of researchers underline the importance of further exploring the implications of participation in e-commerce platforms on export performance in B2B markets [17,52,53]. Therefore, based on the reasoning presented so far, the following hypotheses has been formulated:

**Hypothesis 6:** 
*Presence in marketplaces has a positive and significant relationship with food exports of companies in the B2B context.*


In short, the study addresses a set of quantitative variables (volume of exports, number of publications on the Internet, reviews on social networks and frequency of participation in marketplaces), as well as a set of qualitative variables (online reputation, positive and negative connotations of the publications—sentiments—and quality of the website) in the B2B context, which aim to contribute to understanding the development of the online reputation of food exports companies in e-commerce contexts.

## 3. Materials and Methods

This study focused on Peruvian agri-food exporting companies participating in international trade shows. It should be noted that in Peru’s economy, the agri-food sector has a fundamental role, currently representing almost 25% of the country’s total exports, which go to 142 markets [67].

Previous research indicates that trade shows are highly relevant scenarios for promoting the reputation of participating companies. In fact, participation in B2B fairs frequently has a positive influence on the marketing strategy of companies, given that it is the moment in which strategic information is obtained for exhibitors [68], hence, the great importance of participating in these fairs, which is largely conditioned by the reputation of the company [69]. 

The research was carried out in three stages, in the context of B2B commercial relationships. Figure 1 provides a graphical representation and overview of the entire process.

The first phase of the study adopted a constructive research perspective, integrating the analysis of business issues with the theoretical background [70]. Specifically, the literature review carried out has made it possible to draw up a preliminary list of key elements that make up online reputation, which has been completed and validated with the collaboration of 9 experts from Export Promotion Agencies, Trade Associations and International Organizations from Germany, Japan, Italy, China, Belgium, Brazil, Costa Rica, and Peru. This collaboration took place during the development of 3 international food trade shows: the Cocoa and Chocolate Show and Expoalimentaria in Peru, and Anuga in Germany, in July 2021, September 2021, and October 2021 respectively. These trade shows were usually held in a face-to-face format, but during the COVID-19 period had to be developed in a hybrid and virtual form.

The opinions of the experts participating in the study were obtained through in-depth interviews conducted virtually, due to security measures during the COVID-19 period. The conversations revolved around the aspects they considered decisive for the online reputation of agri-food exporting companies. Experts evaluate the preliminary list of factors developed from the literature review using a 7-point scale (1 = minimum importance; 7 = maximum importance), which, in their opinion, provided them with an appropriate reference framework for expressing and grading their views, and which is commonly accepted in the field of online reputation research [2,21,63]. The items selected for inclusion in the final list were those that scored “high” or “very high” on the scale, which is equivalent to an average score of more than 5 points. In addition, during the interviews, experts provided insights and indications that were taken into account.

To confirm the conceptual structure obtained, as well as to obtain a practical business perspective, 30 managers of Peruvian food companies (14 women and 16 men, with top positions in the export and commercial management of their companies) were interviewed in the context of 5 international trade shows throughout 2021 and 2022. Three of them (virtual format) were the same ones where experts from official institutions and associations were interviewed: Cocoa and Chocolate Show (Peru, July 2021), Expoalimentaria (Peru, September 2021), and Anuga (Germany, October 2021). The two remaining trade shows were Cacao y Chocolate (Peru, July 2022) and Biofach (Germany, July 2022); in both cases, the interviews were conducted in a face-to-face format. The companies participating in the study ranked highest in terms of turnover and exports in Peru, and traded products in the following categories: fruits and vegetables, juices and nectars, cocoa, grain, pulses, flours, spices, pulses, and coffee.

As the experts had previously, the managers provided their views on the most important determinants of the online reputation of their companies and evaluate the list of proposed factors using the same 7-point scale. During the interviews, their questions were responded to, and their comments were taken into account. This stage made it possible to identify the main determinants of the online reputation from the experience of the companies participating in the study. Specifically, 13 elements were identified, grouped into 3 sets:▪Set 1: Website (number of results in the Google search engine with the name of the company; ease of finding the page—SEO positioning in Google (page in the first results); prominence—number of visitors to the website; frequency of content updates; quality of the website; detailed information on products and company (prices, volumes, etc.); storytelling on the website; website translated to English language; brand presence in the digital press).▪Set 2: Social media (presence on social media; influencers who promote the company’s brand and products; quick response to communications and messages in digital media (on social networks, buyer emails, etc.).▪Set 3: Marketplaces (e-commerce channels, B2B marketplaces available for purchases and closing deals).

It should be noted that the methodology used in this first phase of the study incorporates as a novelty with respect to previous research in this area, the consideration of the vision of both experts (perspective from official institutions) and managers (perspective from business practice) of B2B markets.

The second stage of the study was based on the 13 online reputation determinants for food export companies in B2B markets identified in the first stage. A standardized methodology was implemented for the general measurement of online reputation, which was carried out as an extension of the Sentiment Analysis, as explained below. The calculation of the Total value of the Online Reputation (TOR), for each of the food exporting companies participating in this phase of the research, was carried using the methodology of previous studies, in particular, the following formula [71,72,73]:$$TOR=\frac{R_{ASA}+\sum_{i=1}^{n}R_{i}}{n+1}$$
where:TOR—total online reputation in %Ri—reputator (% score based on a given i-th determinant of online reputation) of each food exporter$R_{ASA}$—reputator ASA (% score based on the advanced sentiment analysis)n—number of indicators

Data collection to calculate the TOR was conducted in July and August 2023. Unlike previous studies based on methodologies similar to that of this research, which used a limited sample of companies in assessing online reputation [71,72], this research is based on a larger sample, analyzing 35 Peruvian companies exporting agri-food to Europe. Moreover, this study expands the number of variables to be considered and takes into account aspects of sustainability.

In order to design the instrument for assessing online reputation, the availability of instruments for measuring the determinants identified in the first phase of the study was analyzed, grouping the variables by their method of calculation and sources of information (Table 1). 

It is worth considering that the presence in the digital press through the search in Google news and the results in the Google search engine have been approached with the Advanced Sentiment Analysis (ASA) methodology [71,73], based on the techniques of Sentiment Analysis (SA), which makes it possible to identify the opinions or perceptions of the general public based on text data, useful in the behavioral sciences [74,75].

The polarity of Google search and Google news search results has been used separately, to achieve an extended form of sentiment assessment (Advanced Sentiment Analysis—ASA). Thus, at a qualitative level, each of the posts was evaluated based on a four-state classification: (i) the positive sentiment expressed in the post about the food exporting company; (ii) the position of the company’s website in the search engine; (iii) the neutral sentiment of a post citing the company; and (iv) the negative sentiment expressed in the post about the exporting company. The results of the top 10 positions for each food company were considered, at a quantitative level, using a scoring scheme following the ASA methodology (Table 2). For each of the 10 search results, a food exporting company can achieve a maximum score of 155 points (maximum sum of positive sentiment scores; 1 point = 0.645%) [71,73]. In the extreme case that a food exporting company has negative sentiment in all its publications, it would reach the minimum score of −155 points. In the case where all publications are neutral, the score achieved would be 20 points. In the event that the posts found in the search are unrelated to the company, or the relationship in the search engine is inadequate, a score of zero points will be considered.

The remaining determinants of online reputation were also expressed as a percentage and ranked as follows:Google Score %: based on the number of results with the company name; to express this as a percentage, it is compared to the highest number of results obtained for one of the companies in the sample.Website Score %: based on both quantitative metrics and website characteristics.
$$Website\;Score=\frac{\sum_{i=1}^{n}{Website\;metrics}_{i}+\sum_{i=1}^{n}{Website\;features}_{i}}{n}$$
Website Score: based on the assessment of the website (%).Website metrics: % score based on the number of monthly visits, average duration per visit, and number of pages per visit. After being collected, they have been expressed as a percentage based on the maximum result of each indicator obtained by each company. The bounce rate is already quantified in %. The lowest bounce rate shows a better level of quality of the website, hence the no bounce rate in % has been considered for the score.Website features: % score based on the dummy variables of the existence of a website, sustainability information, products, certifications, and a website in the English language. The dummy variables 0 and 1, expressed in %, are 0% and 100%.n—number of indicators.
Social Media Score %: based on four social networks: Instagram, Facebook, LinkedIn, and YouTube.
$$Social\;Media\;Score=\frac{\sum_{i=1}^{n}{LinkedIn}_{i}+\sum_{i=1}^{n}{Facebook}_{i}+\sum_{i=1}^{n}{Instagram}_{i}+\sum_{i=1}^{n}{Youtube}_{i}}{n}$$
Social Media Score: Score based on the social media platforms assessment by each exporter expressed in %.LinkedIn: % score based on the existence of an official LinkedIn account of the exporter and the number of followers.Facebook: % score based on the existence of an official Facebook account of the exporter, number of followers, likes, mentions, and ratings.Instagram: % score based on the existence of an official Instagram account, number of followers and number of posts.YouTube: % score based on the existence of an official YouTube account of the exporter, number of subscribers and videos.n—number of indicators
Marketplace Score %: based on the presence of the company in any marketplace available on the internet.

As an additional contribution, this research analyses the relationship between the variables that make up online reputation and the exports of the companies participating in the study (regression analysis), which makes it possible to test the hypotheses put forward.

## 4. Results 

### 4.1. First Stage of the Study: Variables That Make up Online Reputation

As described in the section on Materials and Methods, the first phase of the study involved experts from various institutions linked to food exports, as well as managers from leading Peruvian agri-food export companies. This made it possible to identify 13 determining variables for the online reputation of agri-food companies in the B2B market, as well as to obtain the managers’ views on the weighting of each of them (7-point scale). The results obtained are presented In Figure 2 and Figure 3.

As can be observed, the variables considered to be of most importance in shaping online reputation in B2B food markets (with average scores above 6 points) were the quality of the website, detailed information about the products and the company (prices, volumes, etc.), the presence of the brand in the digital press, and the presence in social networks.

In the following range of importance (between 5 and 6 points) were storytelling on the website, e-commerce channels, available B2B marketplaces for purchasing or closing deals, number of visitors to the website, frequency of content updates, correct translation of the website into other languages, ease of finding the website, SEO positioning on Google, and quick response to communications.

Finally, with scores below 5 points were the number of results in the Google search engine with the company’s name and the number of influencers promoting the company’s brand and products.

### 4.2. Second Stage of the Study: Measurement of Total Online Reputation—TOR

The data collected for the 35 companies considered in this second phase of the investigation allowed for the calculation of their respective TOR score.

Table 3 presents the scores for each of the sentiments expressed in the Google search engine and Google News by company evaluated ($R_{\mathrm{ASA}}$—reputator ASA). The results are expressed in percentage based on a maximum of 155 points.

The TOR score (%) has been obtained on the basis of the average of the individual determinants of reputation obtained between July and August 2023. Table 4 shows each of the components that are part of the TOR developed for exporting companies in the food sector: ASA Score, Google Score, Website Score, Social Media Score, and Marketplace Score (measures of these determinants of online reputation are detailed in the Materials and Methods section).

It is important to note that the Social Media Score shows a similar trend to the TOR Score. However, the ASA Score does not show a similar evolution to the TOR Score, even being negative in one of the cases of the study for the top 10 exporting companies (Figure 4).

For those 10 companies with the highest export volumes, a more detailed analysis of their ASA score for the first pages indexed in the search engine has also been developed. Specifically, the ASA score has been measured not only globally, but also disaggregated into its two components (ASA Google search and ASA Google news). The results show that the overall ASA score and the ASA Google search and ASA Google news scores have a similar trend (Figure 5).

Finally, a comparison was made between the export volume of the 10 companies considered and their respective social media scores. Specifically, two social networks (LinkedIn and Facebook) were considered, as they had the highest number of companies with available accounts. As can be seen in Figure 6, there is a notable difference depending on the social network. This is most likely due to the different strategies developed by companies in this area.

These differences lead to the need for further analysis using a regression model, which is presented below.

### 4.3. Third Stage of the Study: Regression Model for Impact on Exports

The third phase of the study aimed to assess the relationship between the volume of exports and the TOR score of the 35 companies analyzed in the second phase of the study. Specifically, exports per company for the year 2020 were considered as the dependent variable, and the TOR score as the independent variable. In addition, this relationship has also been analyzed for each of the components of the TOR (online reputation determinants), always on an individual basis, i.e., considering a single independent variable (simple linear regression). The results obtained are presented in Table 5.

The results obtained lead to the acceptance of most of the hypotheses put forward in that study. In particular, it is observed that the total online reputation (TOR), the volume of publications related to the exporting company indexed on the internet (Google Search Score), the quality of the website (website reputation), and the presence on social networks (social media reputation) as well as in marketplaces (marketplace score), has a significant relationship with the export volume of food companies in the B2B context.

In the case of total ASA, no significant influence on exports is obtained. This is explained by one of its components, the ASA Google news, given that many of the exporting companies did not have news in the Google search engine, especially the SMEs. However, there was a positive relationship in the case of ASA Google search (*p*-value ANOVA > 0.05), although with a low R-squared (13.6%) value.

This is followed by Table 6, which gives an overview of the hypotheses accepted and rejected in the study.

The results also show that the R-squared of the TOR as an independent variable explaining exports stands at 51%. This value is significant, but should be taken with caution, linked to the context of this case of Peruvian companies exporting food to Europe. Nevertheless, it gives important indications that may suggest a significant relationship. 

On the other hand, web reputation, measured by quantitative and qualitative variables (number of monthly visits, average duration per visit, number of pages per visit, existence of a website, sustainability and product information, certifications, and a website in the English language) reaches 58.2% significance in explaining exports, while reputation based on social networks (social media platform assessment) achieves 41.9%, and Google indexing 33.7%. Despite information limitations, the company’s presence in marketplaces scores 24.1%.

In order to identify which variables within these reputation determinants have a significant relationship with the volume of exports, a multiple linear regression was performed (Table 7).

As can be observed, the two variables with the highest R-squared values are the number of pages consulted per visit to the company’s website, and the dummy variables for sustainability information on the website (in this study, the variable “sustainability” refers to whether the company has public information on its website about the company’s sustainability activities). Likewise, the certifications reach an outstanding value. Two alternatives were also planned with social media reputation and ASA Google search, but the t-values of these variables were close to or slightly higher than 0.05, so they should be considered with caution. None of the other variables in the study generated significant or relevant results. 

## 5. Discussion

The online reputation of companies turns out to be crucial for increasing competitiveness in business transactions. This is demonstrated by previous studies in areas as diverse as family businesses [73], the banking sector [76], the tourism sector [41,64], and the textile industry [50,77], among others. Therefore, it is vital to understand the key factors that determine the online reputation, incorporating in the analysis both the perspective of the companies and that of their stakeholders [78].

This study focuses on the food export sector in B2B markets, an area of major importance in the economy of many. The results confirm, at a conceptual level, the findings of previous research on the importance of reputation as an intangible asset for the competitiveness and profitability of food companies [54,79], insofar as it predisposes and conditions customer evaluations. Thus, firms with favorable reputations are more likely to achieve higher levels of value associated with their products [80].

More specifically, with respect to the hypotheses put forward, the overall online reputation shows a significant relationship with the export volume of food companies (hypothesis 1 is confirmed), with TOR indicator reaching an explanatory value of 51%. This result is in line with research in the area of exports of food products [17,18], which highlights the key role of reputation in the internationalization of the sector [15,29]. The added value of this research is that it focuses specifically on online reputation for food exporting companies in B2B markets, since most of studies address the issue in B2C markets [20].

In order to gain a more detailed understanding of the influence that the online reputation of food companies can exert on their exports, the study has investigated the aspects that make up this online reputation, providing an overview that is one of its main contributions. Specifically, interviews with experts and managers have led to the identification of three groups of key factors linked to the company’s website, its social media management, and its e-marketplace presence.

With regard to the company’s website, the results indicate that it covers both quantitative aspects (number of publications in the Google search engine, positioning of the company in the top search results, number of visitors to the website, frequency of content updates, and presence in the digital press), and qualitative features of the website (detailed information on products and prices, on the company’s sustainability activities and on official certifications obtained, storytelling on the website and translation of the website into English).

These results are supported by previous research in B2C markets. For example, Kim and Lennon (2013) [55] linked online reputation to the quality of the website and the emotions experienced by the customers who use it, Vollrath and Villegas (2022) [33] pointed out the determinant role of the reliability of the information published, Thongmak (2022) [81] highlighted the usability and security of the website, and Roy and Sharma (2021) [82] corroborated the impact of average session duration and repeat visits on consumers’ online behavior. Therefore, the website determinants of online reputation in B2B and B2C markets show strong similarities.

The study also made it possible to verify the extent to which these quantitative and qualitative characteristics of the websites of the food companies participating in the study influence their exports in B2B markets, through regression analyses carried out.

Regarding the volume of publications linked to each exporting company (the number of results in search engines with the company name), the results obtained indicate that they are significantly related to the company’s export volume. Hypothesis 2 is therefore confirmed. This result corroborates the importance, already detected in B2C markets, of the online presence and visibility of food exporting companies in order to generate an adequate reputation and achieve successful commercial strategies [3,40] through notoriety [1,36]. It should be noted, however, that the explanatory power of the Google Search Score (based on the results with the company name) reaches a moderate value of 33.7%. This is to some extent in line with the view of the managers who participated in the first phase of the study, who placed this element in a less prominent position of importance.

With regard to the feelings generated by these publications, the sentiment analysis carried out under Google news and Google search does not reveal significant effects on exports. Therefore, hypotheses 3 is not confirmed. This result contrasts with those of studies in which sentiment analysis was presented as relevant for information published with regard to financial data [83,84] and CSR reports [37,85] of companies operating in B2C markets. As reasons, the difference between the B2B and the B2C contexts can be pointed out, the latter generating a greater number of publications [86], and companies being more exposed to public information [22,87]. In addition, customer profiles vary in both types of markets, with B2B markets characterized by transactions carried out by specialized committees that make professional assessments of available information [35,53].

Interestingly, the results show that, in the context of online B2B food exports, it is the quality of the publications (website reputation), rather than the quantity of publications or the sentiment they generate, that is relevant. This represents one of the main findings of the study. Thus, the quality of the website, including reliable sustainability information and certifications, shows a significant relationship with the volume of exports of the companies (hypothesis 4 is confirmed). Overall, the regression model shows that web reputation explains 58.2% of the exports of the food companies analyzed. This finding supports the importance for food exporting companies of strengthening their online reputation through the technical information they communicate to their client companies using their websites, especially information on the quality of their products and processes (certifications) and their sustainability practices linked to food systems [48,54]. It should also be pointed out that it was the factors linked to the quality of the website that the managers who participated in the first phase of the study placed first in terms of importance.

The results of the study are reinforced by previous research highlighting the relevance of website quality in achieving the objectives of market positioning [88], increasing market value of firms [81] and fostering exports [89]. They also confirm the importance of providing public information on sustainability on websites and the positive effect this has on companies’ reputation [90,91]. 

Moreover, remarkably, multiple regression models in which variables related to web reputation are integrated (more precisely, presence of sustainability information and number of page views per visit), achieve 73.1% explanation of exports for the sample of this research. Alternatively, a model considering information on certifications instead of sustainability reaches 71.5% explanation. Therefore, the number of pages viewed on each visit seems more relevant in relation to food exports than the number of overall visitors or pages indexed in Google.

Therefore, the number of pages viewed on each visit seems more relevant in relation to food exports than the number of overall visitors or pages indexed in Google. This result has great relevance, given that companies with higher levels of exports would have visitors who search for more information on each visit, feeling more interested in reviewing larger parts of the company’s website. This also suggests that companies in the food sector should have a storytelling on their website that leads to a greater number of pages that are visited, in which information on sustainability and certifications plays an important role. The relevance of measuring the number of pages per visit is theoretically supported by previous studies, which also delve into aspects such as the average duration of the session and repeat visits [82], as well as the intensity of the interaction and the level of engagement generated [92].

The relationship between the presence in social networks and food exports is also relevant in this research (hypothesis H5 is confirmed), although with somewhat less importance than that of the website of the exporting companies. In particular, social media reputation explains 41.9% of exports, compared to 58.2% for website reputation. This may be due to the fact that, unlike in the B2C context, where social networks are more established and developed, in the B2B context of food exports, international buyers do not consider them as relevant as sources of information for deciding on a purchase [34]. Nor do companies tend to consider them as the most appropriate way to engage with their customers [35]. Despite this, further analysis is required, in order to enhance the use of social media in B2B markets and reinforce the brand image of companies, create attractive content, and even train and motivate employees in their interactions with client companies [56]. This would be the case of LinkedIn and Facebook in our study, as they are the two social networks with the highest number of accounts for the companies analyzed.

The results also show that social networks, in combination with the company’s website, constitute a crucial online strategy in generating the online reputation of food exporting companies, reaching in this study jointly a 70% explanation of B2B exports. This approach is supported by the results of studies conducted in the context of importing and exporting food products, as well as the role played by social media in the attitudes and food behaviors of users. For instance, during the COVID-19 crisis, the health safety of imported food became highly relevant on social media, which turned into a key medium for providing information, reducing uncertainty, and channeling customers’ emotions [87]. Likewise, social networks, combined with company websites, become decisive in the promotion of eating habits and the acceptance of new foods [86].

Finally, presence in marketplaces shows a significant relationship with food exports (hypothesis 6 is confirmed), although with less explanatory capacity for food exports (24.1%) than the company’s website (58.2%) and social media (41.9%). These results contrast with previous theory and studies on the positive importance of marketplaces for B2C business. In fact, e-commerce channels currently play a strategic role in B2C markets, becoming key parts of companies’ commercial strategies to generate information flows [41], reinforcing the positioning of products and brands [64,93], and increasing efficiency by reducing costs [17].

The main reasons for the slower development of marketplaces in B2B food exports include, according to previous studies on the topic, the closer and more direct relationships that tend to be established between selling and buying companies [44,65], given the large volumes of trading in this type of business, as well as the direct control that buying companies tend to exercise over the quality and safety of foodstuffs [66]. 

In summary, the study has confirmed five of the six hypotheses put forward in the research (all of them except the one were related to the sentiment of food exporting companies’ publications on the internet) and highlights the need to adapt the different aspects that determine the online reputation of companies to the particularities of B2B markets.

## 6. Conclusions

The research conducted provides a number of implications for the online reputation management of food exporting companies in B2B markets.

From an academic point of view, the main components of online reputation in the sector have been identified, using a holistic methodology that takes into account the contributions of other researchers in the field, the vision of the heads of large official institutions linked to exports, as well as the reality of business practice. In addition, the study proposes an adapted index (TOR) to assess the online reputation in B2B markets and confirms the significant influence of online reputation on export performance. Therefore, the research provides conceptual tools that can also serve as a guide for companies in the sector, adapted to the particularities of B2B markets, thus contributing to the advancement of the theory and practical management of online reputation in the food context.

From a managerial perspective, this study highlights the fundamental aspects of online reputation that significantly condition the volume of food exports: the quality and information of the website, the presence in social networks and the availability of e-commerce channels in marketplaces.

More specifically, companies in the sector should pay special attention to incorporating relevant and reliable information on their websites, prioritizing the sustainability aspects of their activities and strategies, as well as the certification of their products and processes. As far as social media is concerned, their role as promoters of food exports needs to be considered by food exporting companies. It is not just a matter of being present on social media, but of selecting those that are most appropriate for the positioning of the company and for the profile of its customers in B2B contexts. Finally, the presence in online marketplaces should not be neglected either. In this case, the challenge is to offer agile and efficient channels that foster e-markets in B2B contexts, with up-to-date information on products, prices, and other factors that can be decisive for the success of exports.

In conclusion, the proper management of the website, the social media, and the marketplace of food exporting companies represents a strategic opportunity to promote their online reputation among their client companies, impacting both the attitudes and behaviors of these clients, as well as the final consumers to whom they will deliver their products.

This information is also of considerable interest to official organizations and agencies in food export fields, given the crucial importance of this sector in the economy of many countries, as is the case in Peru, the country on which the empirical study was based. These official entities have the possibility of supporting public opinion [87], and of strengthening the reputation of their food sector, reinforcing those aspects that are most decisive in fostering their export through B2B markets [18,19].

The results stimulate future directions in this field of research. On the one hand, the study provides information on the components of online reputation from an academic and business perspective. It would also be of great interest to know what elements are relevant to customer companies in generating their perceptions of the online reputation of vendor companies. On the other hand, future studies focusing on customer companies can provide precise ideas on their information needs regarding the companies’ websites, the detailed information they find most relevant in their purchasing decisions, the formats of the information they trust the most, and the type of information and communication codes that generate the most favorable sentiments towards the selling companies.

The results obtained have not confirmed that the sentiment generated by company publications significantly enhances their exports. Nevertheless, the pursuit of continuous improvement by companies working in B2B markets encourages research into the sentiments generated by their publications as sources of competitive advantage [31].

It is also of great interest to explore the subject of social networks to promote B2B food exports, identifying those that may be more appropriate, as well as formats that are more adapted to the sector. For instance, it is of interest to know what types of information are more suitable to be shared on social media in this context, as compared to the website, what communication codes are the most appropriate, and what type of figures can play the role of influencer.

Finally, the field of marketplaces in B2B contexts remains largely unexplored, opening the way for future research [17,53]. In this case, the aim is to develop studies that guide their use as efficient supports for making purchases and closing deals, and also as an effective means of communication between the different stakeholders involved in the organizational purchasing process [94].

This research has limitations, which serve as a stimulus for additional studies in this relevant and under-addressed area. With regard to the not very relevant result obtained when analyzing the influence of marketplaces on food exports, the scarce information available in this study in this respect should be noted, which means that more in-depth and diversified information is required to be able to contrast this finding. It would also be highly recommendable to carry out longitudinal studies to observe the evolution over time of online reputation indicators, as well as their relationship with food company exports. Finally, it should be noted that the study was based on a sample of food exports companies in Peru, which operate in a specific economic and social context. It is therefore necessary to reproduce this type of study in other economic spheres, with companies from other countries, in order to establish a comparison of the results obtained and to achieve a more global perspective.

## Figures and Tables

**Figure 1 foods-12-03862-f001:**
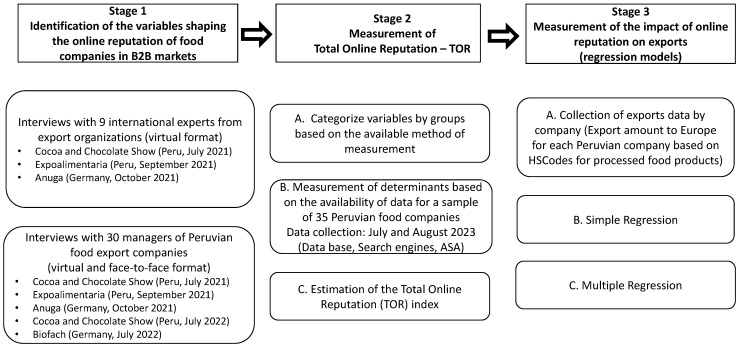
Phases of the study.

**Figure 2 foods-12-03862-f002:**
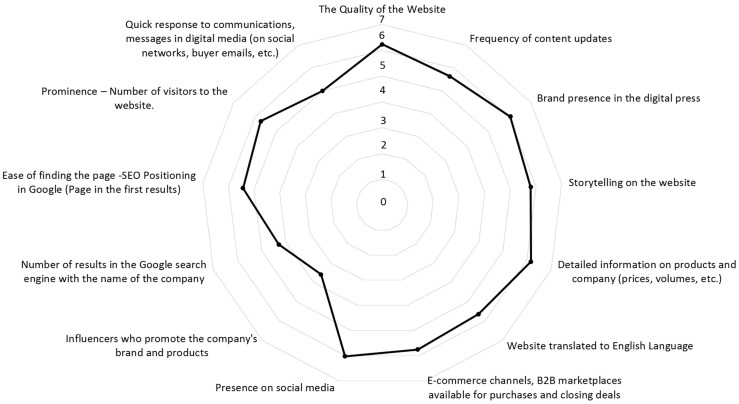
Variables that integrate B2B digital reputation and average importance.

**Figure 3 foods-12-03862-f003:**
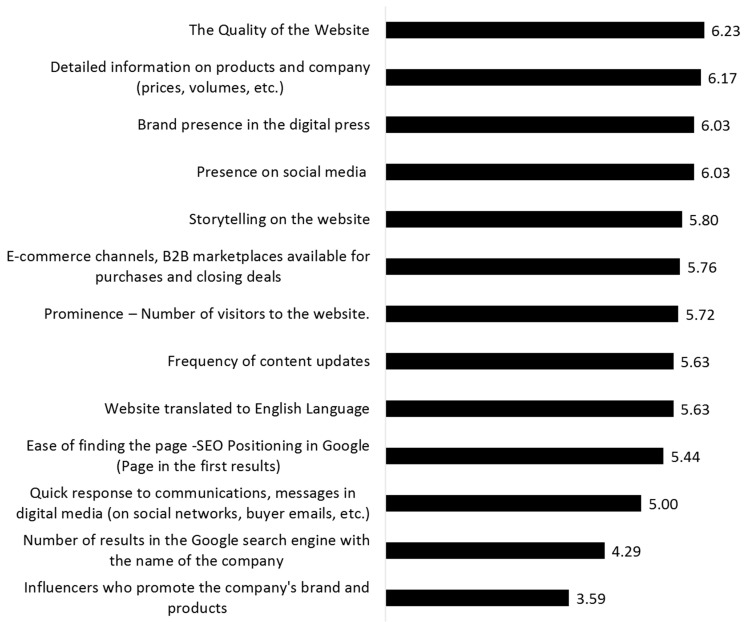
Ranking of digital reputation variables in the study.

**Figure 4 foods-12-03862-f004:**
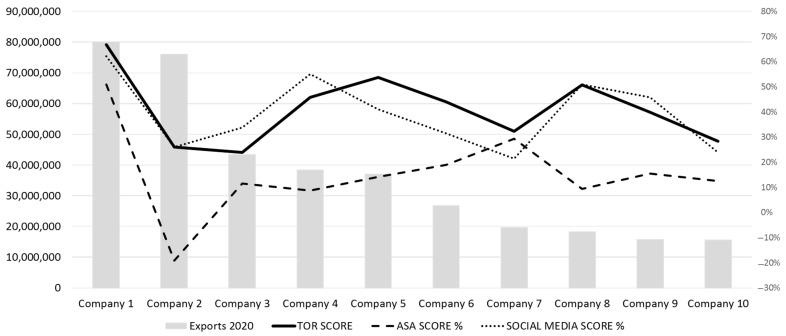
Exports vs. TOR Score, ASA Score, and Social Media Score for the 10 largest companies in the sample.

**Figure 5 foods-12-03862-f005:**
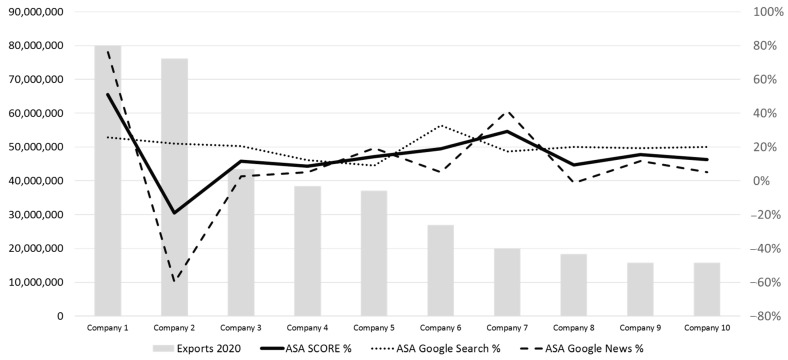
Exports vs. ASA Score total score and two components: ASA Google search and ASA Google news.

**Figure 6 foods-12-03862-f006:**
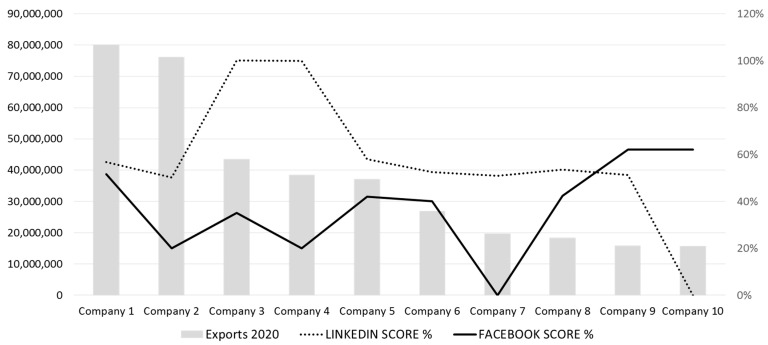
Exports vs. LinkedIn Score and Facebook Score.

**Table 1 foods-12-03862-t001:** Measurement of the determinants of online reputation of food exporting companies in B2B markets.

Category	Reputation Determinants (Survey)	Measuring Mechanism	Information Source/Calculation/Nature
I. WEBSITE	Number of results in the Google search engine with the name of the company	Number of results in search engines	Google search (quantitative)
	Ease of finding the page—SEO positioning in Google (page in the first results) Brand presence in the digital press	2. Advanced Sentiment Analysis (ASA)	ASA methodology using the first 10 results of Google search and Google news (qualitative/quantitative)
	Prominence—number of visitors to the website Frequency of content updates	Number of monthly visits Average duration per visit Number of pages per visit Bounce rate	SimilarWeb: Website traffic (quantitative)
I. WEBSITE	The quality of the website Detailed information on products and company (prices, volumes, etc.) Storytelling on the website Website translated into the English language	5. Dummy variable whether the company has a website 6. Dummy variable if the company has information about the sustainability of the company 7. Dummy variable if there is information on the company’s products 8. Dummy variable if the company has information about the certifications of the company’s products 9. Dummy variable whether the company has a website in English	1 = has information on the web, 0 = does not have information on the web (qualitative)
II. SOCIAL MEDIA	Presence on social media ^1^ Influencers who promote the company’s brand and products Quick response to communications, messages in digital media (on social networks, buyer emails, etc.)	LinkedIn 10. Number of followers on LinkedIn 11. Dummy variable whether the company has LinkedIn Facebook: 12. Number of followers on Facebook 13. Dummy variable whether the company has an official Facebook account 14. Number of likes on Facebook 15. Number of mentions on Facebook 16. Rating Opinion Score on Facebook Instagram: 17. Number of followers on Instagram 18. Number of posts on Instagram 19. Dummy variable whether the company has an official Instagram account Youtube: 20. Dummy variable if the company has or does not have a YouTube account 21. Number of subscribers on YouTube 22. Number of videos	Information of the company’s LinkedIn, Facebook, Instagram, and YouTube (Quantitative)
III. MARKETPLACES	E-commerce channels, B2B marketplaces available for purchases and closing deals	23. Dummy variable if the company has or presence in a marketplace online with prices of their products	It has a presence of its products with prices in some marketplaces = 1, it does not have a presence = 0 (qualitative)

^1^ Twitter (currently called X) was not included due to the very limited number of Peruvian exporters having accounts on this platform.

**Table 2 foods-12-03862-t002:** Sentiment analysis quantification.

Position of the Result	1	2	3	4	5	6	7	8	9	10	Max
Sentiment											
Positive sentiment (+)	20	19	18	17	16	15	14	13	12	11	155
Custom website of the organization (x)	10	9	8	7	6	5	4	3	2	1	55
Neutral sentiment (±)	2	2	2	2	2	2	2	2	2	2	20
Negative sentiment (−)	−20	−19	−18	−17	−16	−15	−14	−13	−12	−11	−155

**Table 3 foods-12-03862-t003:** Advanced Sentiment analysis by company.

Company	ASA Google Search	ASA Google Search (%)	ASA Google News	ASA Google News (%)	ASA SCORE (%)
Company 1	40	26%	118	76%	51%
Company 2	34	22%	−93	−60%	−19%
Company 3	32	21%	4	3%	12%
Company 4	19	12%	8	5%	9%
Company 5	14	9%	30	19%	14%
Company 6	51	33%	8	5%	19%
Company 7	27	17%	64	41%	29%
Company 8	31	20%	−2	−1%	9%
Company 9	30	19%	18	12%	15%
Company 10	31	20%	8	5%	13%
Company 11	28	18%	71	46%	32%
Company 12	16	10%	10	6%	8%
Company 13	67	43%	0	0%	22%
Company 14	44	28%	6	4%	16%
Company 15	33	21%	2	1%	11%
Company 16	28	18%	6	4%	11%
Company 17	31	20%	0	0%	10%
Company 18	16	10%	0	0%	5%
Company 19	6	4%	0	0%	2%
Company 20	12	8%	0	0%	4%
Company 21	22	14%	0	0%	7%
Company 22	13	8%	4	3%	5%
Company 23	33	21%	6	4%	13%
Company 24	8	5%	0	0%	3%
Company 25	2	1%	0	0%	1%
Company 26	19	12%	0	0%	6%
Company 27	2	1%	6	4%	3%
Company 28	8	5%	0	0%	3%
Company 29	10	6%	0	0%	3%
Company 30	16	10%	0	0%	5%
Company 31	24	15%	0	0%	8%
Company 32	0	0%	0	0%	0%
Company 33	10	6%	0	0%	3%
Company 34	6	4%	0	0%	2%
Company 35	2	1%	0	0%	1%

**Table 4 foods-12-03862-t004:** Results of calculation of reputation determinants and total TOR score.

Company	ASA Score (%)	Google Score (%)	Website Score (%)	Social Media Score (%)	Marketplace Score (%)	TOR Score (%)
Company 1	51%	80%	71%	62%	100%	67%
Company 2	−19%	7%	63%	26%	100%	26%
Company 3	12%	8%	49%	34%	0%	24%
Company 4	9%	13%	62%	55%	100%	46%
Company 5	14%	100%	91%	41%	100%	54%
Company 6	19%	74%	58%	32%	100%	44%
Company 7	29%	8%	37%	22%	100%	32%
Company 8	9%	75%	59%	51%	100%	51%
Company 9	15%	8%	36%	46%	100%	40%
Company 10	13%	1%	33%	24%	100%	28%
Company 11	32%	4%	46%	42%	0%	31%
Company 12	8%	0%	0%	5%	0%	4%
Company 13	22%	1%	25%	0%	0%	8%
Company 14	16%	1%	21%	0%	100%	17%
Company 15	11%	1%	25%	0%	0%	5%
Company 16	11%	2%	25%	18%	100%	24%
Company 17	10%	1%	30%	0%	0%	6%
Company 18	5%	0%	0%	0%	0%	1%
Company 19	2%	0%	0%	0%	100%	12%
Company 20	4%	0%	0%	0%	0%	1%
Company 21	7%	0%	25%	32%	0%	18%
Company 22	5%	0%	25%	13%	0%	10%
Company 23	13%	0%	21%	27%	0%	17%
Company 24	3%	0%	0%	0%	0%	1%
Company 25	1%	0%	0%	0%	0%	0%
Company 26	6%	0%	46%	0%	0%	6%
Company 27	3%	0%	0%	0%	0%	1%
Company 28	3%	0%	0%	0%	100%	12%
Company 29	3%	0%	0%	0%	0%	1%
Company 30	5%	0%	0%	0%	0%	1%
Company 31	8%	0%	0%	57%	100%	38%
Company 32	0%	0%	0%	0%	0%	0%
Company 33	3%	0%	29%	0%	0%	4%
Company 34	2%	1%	0%	0%	0%	1%
Company 35	1%	0%	0%	0%	0%	0%

**Table 5 foods-12-03862-t005:** Simple linear regression results: Exports vs. TOR and online reputation determinants.

N	Independent Variable	R	R-Squared	Adjusted R-Squared	ANOVA Sig
1	TOR	0.714	0.510	0.496	0.000
2	ASA total	0.310	0.096	0.069	0.070
3	ASA Google search	0.369	0.136	0.110	0.029
4	ASA Google news	0.180	0.032	0.03	0.302
5	Google Search score	0.581	0.337	0.317	0.000
6	Website reputation	0.763	0.582	0.569	0.000
7	Social media reputation	0.647	0.419	0.401	0.000
8	Marketplace score	0.491	0.241	0.218	0.003

**Table 6 foods-12-03862-t006:** Results for the hypotheses formulated in the study.

Hypothesis	Result
Hypothesis 1: online reputation has a significant relationship with the volume of food exports of companies in the B2B context.	Accepted
Hypothesis 2: the volume of publications related to the exporting company indexed on the internet has a significant relationship with the volume of exports.	Accepted
Hypothesis 3: the sentiment of food exporting companies’ publications indexed on the internet has a significant relationship with their export volume.	Not accepted (ASA total) Not accepted (ASA Google news) Accepted (ASA Google search)
Hypothesis 4: the quality of the website, including sustainability information and certifications, has a positive relationship with the export volume of exporting companies in the B2B context.	Accepted
Hypothesis 5: presence in social networks has a positive and significant relationship with food exports of companies in the B2B context.	Accepted
Hypothesis 6: presence in marketplaces has a positive and significant relationship with food exports of companies in the B2B context.	Accepted

**Table 7 foods-12-03862-t007:** Multiple linear regression results.

N	Independent Variable	R	R-Squared	Adjusted R-Squared	ANOVA Sig	t	Tolerance	VIF
1	(a) Sustainability (b) Pages per visit	0.855	0.731	0.714	0.000	0.007 0.000	0.562	1.779
2	(a) Certifications (b) Pages per visit	0.846	0.715	0.697	0.000	0.018 0.000	0.622	1.607
3	(a) Social media reputation (b) Pages per visit	0.837	0.700	0.682	0.000	0.046 0.000	0.642	1.557
4	(a) ASA Google search (b) Pages per visit	0.834	0.696	0.677	0.000	0.060 0.000	0.949	1.054

## Data Availability

Data are contained within the article and available upon request.

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
