# Peer review of "Online Reputation for Food Sector Exporters in the B2B Context: The Importance of Sustainability"

_foods, 2023, doi:10.3390/foods12203862_

Round 1
Reviewer 1 Report
I can state that the topic of the article is interesting.
The abstract is relatively well structured, specifies the issue, incl. research gap, defines the aim of the authors, and presents the findings. Please, specify used methodology.
The introduction section is not well structured and argued. Please, define the issue, explain why the given issue is topical and important, specify research gap and aim, and summarize the findings on the given topic to date.
The theoretical background is not sufficiently elaborated. The hypotheses are not directly based on the findings of the literature review.
The Materials and Methods section should be improved. The methodology needs to be accurately described to be replicable. It is also unclear whether the methodology was adopted or created by the authors. The period of research implementation is not specified. Even this section needs to be supported by the works of other authors.
Regarding the "Results" section, the findings are not clearly presented. Was it interviews or a questionnaire? In the case of the questionnaire, the number of 35 respondents is not sufficient.
As for the discussion, avoid mass citations, e.g. [27,40,75,76, 77,78,79] or [58,67, 68,69,61] etc. Conduct a more in-depth discussion with respect to individual, previously published works.
It can be stated that the conclusions are appropriately processed.
Reviewer 2 Report
The paper is topical and well-structured. Following are my concerns
1. In the abstract section the author must highlight the implications of the research.
2. The development of hypotheses (line 126-143) need more rigor while justifying how are they grounded on past studies and extending the volume of literature.
3. Line 163-164: Why have you selected 30 managers? It is better to give their profile.
4. Why have you taken up 7 point scale? Needs justification
5. Figure captions normally are given below the figures.
6. Critical analysis of the results (not description) is missing
7. Add to future directions
Moderate editing is required.
Reviewer 3 Report
Thanks for this timely and interesting manuscript entitled “ Online reputation for food sector exporters in the B2B context: the importance of Sustainability”. Overall the manuscript is interesting and adds to the knowledge. The authors have undertook a mix of qualitative and quantitative approach to enrich their study and achieve its objectives. The goals of the study and its significant is well-presented in the introduction. Despite the authors have spent good efforts in preparing this manuscript. There are some part of the manuscript which needs more effort. Here are the points that needs more clarification please:
· Definitions in social sciences have often little consensus as each scholar has his/her argument and justification of the definition. Please cite some of these definitions concerning the online reputation in the introduction and how it is measured by other scholars (even in B2C business).
· It was not clear what is the theoretical framework adopted in this study. It is unclear which theory was adopted for undertraining the online reputation. Please add this to the theoretical background of the study.
· I think that the theoretical background need more efforts. You have hypothesized 6 hypotheses but did not provide enough studies that support your arguments. Fro example none of the studies were scited to support you hypotheses 5 and 6. You need to consider more studies even in the context of B2C or other related studies. See for instance:
Wilis, R.A.; Nurwulandari, A. The effect of E-Service Quality, E-Trust, Price and Brand Image Towards E-Satisfaction and Its Impact on E-Loyalty of Traveloka’s Customer. J. Ilm. MEA 2020, 4, 1061–1099.
Alnaim, A.F.; Sobaih, A.E.E.; Elshaer, I.A. Measuring the Mediating Roles of E-Trust and E-Satisfaction in the Relationship between E-Service Quality and E-Loyalty: A Structural Modeling Approach. Mathematics 2022, 10, 2328. https://doi.org/10.3390/math10132328
· I suggest you summarize the steps of the methods undertaken in your study in a figure or a diagram to be easier for understanding.
· It is unclear for me how did you measure “sustainability” as independent variable in Table 5.
· I suggest you add a table in the results or discussion section to confirm or reject your hypotheses highlighted earlier in the literature please.
· I suggest you add implication of your study for the business and scholars as well.
Other minor things that should be considered:
· In some cases you cite several references but it did not show which reference is related to this text such as reference (18-21) line 45 and references 31,32,33,16) in line 100 as an example.
· Please consider revising the sentence started with likewise, considering …….. in line 32. You may consider removing likewise for the beginning of the sentence, as it does not add to the text.
Best whishes
Overall, the English is fine. Minor issues were detected.
Round 2
Reviewer 1 Report
I can state that all my comments have been adequately incorporated. Thank you.
Reviewer 2 Report
I appreciate the effort put up by the authors to revise the manuscript. I don't have any further comment.
Minor typo checking is required
Reviewer 3 Report
Many thanks for this revised version of the manuscript. The authors have spent good efforts in revising their paper. The paper is now more suitable for publication.
Best wishes with your publication